# Locus coeruleus spiking differently correlates with S1 cortex activity and pupil diameter in a tactile detection task

Hongdian Yang[1]*, Bilal A Bari[2], Jeremiah Y Cohen[2], Daniel H O'Connor[2]*

[1]Department of Molecular, Cell and Systems Biology, University of California, Riverside, Riverside, United States; [2]Department of Neuroscience, Brain Science Institute, and Kavli Neuroscience Discovery Institute, Johns Hopkins School of Medicine, Baltimore, United States

**Abstract** We examined the relationships between activity in the locus coeruleus (LC), activity in the primary somatosensory cortex (S1), and pupil diameter in mice performing a tactile detection task. While LC spiking consistently preceded S1 membrane potential depolarization and pupil dilation, the correlation between S1 and pupil was more heterogeneous. Furthermore, the relationships between LC, S1, and pupil varied on timescales of sub-seconds to seconds within trials. Our data suggest that pupil diameter can be dissociated from LC spiking and cannot be used as a stationary index of LC activity.

## Introduction

Multiple lines of evidence implicate the locus coeruleus/norepinephrine (LC/NE) system in perceptual task performance. First, LC activity modulates feedforward processing of sensory stimuli (*Hirata et al., 2006*; *Devilbiss et al., 2006*; *Rodenkirch et al., 2019*) and impacts sensory cortex states (*Constantinople and Bruno, 2011*; *Polack et al., 2013*). Second, LC activity correlates with task performance (*Rajkowski et al., 1994*; *Usher et al., 1999*) and pupil diameter (*Rajkowski et al., 1994*; *Joshi et al., 2016*; *Reimer et al., 2016*; *Liu et al., 2017*). Finally, pupil diameter is thought to index arousal and has been found to be correlated with neuronal and behavioral detection or discrimination sensitivity (*Reimer et al., 2014*; *McGinley et al., 2015a*; *McGinley et al., 2015b*; *Vinck et al., 2015*; *Lee and Margolis, 2016*; *Schriver et al., 2018*; *Lee et al., 2020*; *Cazettes et al., 2021*), as well as decision bias (*de Gee et al., 2014*; *de Gee et al., 2020*). Since sensory cortex activity impacts perceptual reports (*Sachidhanandam et al., 2013*; *Miyashita and Feldman, 2013*), these observations suggest the hypothesis that LC/NE modulates sensory cortex activity and affects perceptual task performance and that this effect can be monitored noninvasively via the easy-to-measure pupil diameter. Testing this hypothesis requires simultaneous measurement of (1) LC activity, (2) cortical activity, ideally subthreshold membrane potential, and (3) pupil diameter, all during perceptual task performance. Here, we recorded spiking activity of optogenetically-tagged LC units together with pupil diameter in mice performing a tactile detection task (*Yang et al., 2016*). In a subset of experiments, we also performed simultaneous whole-cell current clamp recordings in S1 (*Figure 1*).

## Results

First, we report the analysis of LC and pupil recordings during behavior (e.g., *Figure 2a*). Consistent with prior reports (*Joshi et al., 2016*; *Reimer et al., 2016*; *Liu et al., 2017*), cross-correlogram analysis revealed that LC spiking activity and pupil diameter were correlated across entire sessions, with

*For correspondence:
hongdian@ucr.edu (HY);
dan.oconnor@jhmi.edu (DHO)

**Competing interests:** The authors declare that no competing interests exist.

**Figure 1.** Cortical membrane potential, LC spike rate, and pupil recorded during a tactile detection task. (**a**) Task schematic, trial structure, and all trial types of the single-whisker detection task (*Yang et al., 2016*). (**b**) Schematic of tetrode recording in LC, whole-cell recording in S1, and pupil tracking during the task. (**c**) Expression of ChR2 in a Dbh;Ai32 mouse (ChR2-EYFP: green; tyrosine hydroxylase TH: red). (**d**) Left: Responses of a ChR2-expressing LC unit to opto-tagging (lightning bolts: blue light pulses) and tail pinch. Middle: LC unit responses to 12 blue light pulses (200 ms) aligned to individual pulse onset. Ticks represent spikes. PSTH is shown at the bottom. Right: Typical wide waveforms of LC units and an electrolytic lesion (arrow: lesion site) in the LC (white) showing the recording location. (**e**) Example simultaneously recorded LC activity, S1 $V_m$, and pupil with auditory cue and whisker stimulation onsets indicated. Trace is from a brief period of non-performance during a behavioral session and so there are no licks. The example pupil size is typical for all sessions (*Figure 1—figure supplement 1*).

The online version of this article includes the following source data and figure supplement(s) for figure 1:

**Figure supplement 1.** Pupil diameter across recordings.

**Figure supplement 1—source data 1.** MATLAB R2016b file with median pupil diameter data.

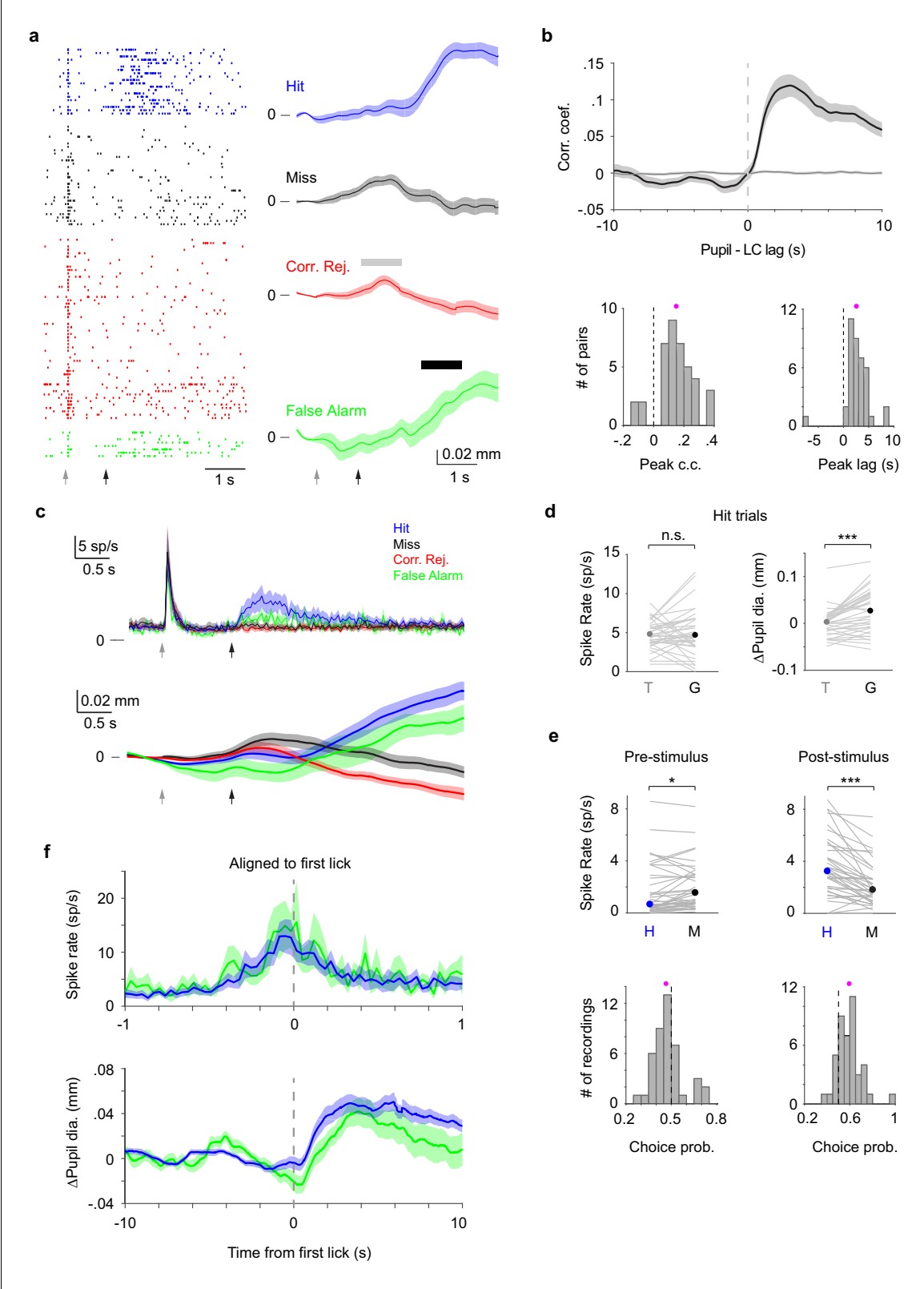

**Figure 2.** LC and pupil responses during behavior. (a) Example LC recording with pupil tracking. Left: LC spike raster separated by trial types. Right: Mean pupil diameter (± s.e.m.) separated by trial types. Gray and black arrows indicate tone and stimulus onsets, respectively. Gray and black bars indicate the time windows during which pupil responses to tone and to Go (behavioral responses) were quantified, respectively. We note that based on the temporal profiles of pupil diameter in different trial types (i.e., in the presence or absence of tactile stimulus or licking) and that the tactile stimulus

*Figure 2 continued on next page*

*Figure 2 continued*

starts 1 s after tone onset, pupil responses to tone and Go can be segregated (Materials and methods). (**b**) Top: Grand average cross-correlogram between LC spike train and pupil diameter (n = 39). Individual LC spikes were convolved with a 400 ms wide Gaussian kernel. Spike times were shuffled and LC–pupil correlations computed to establish controls (narrow gray band around zero). Bottom: Histogram of peak correlation coefficient (left), and time lags (right) between LC spike train and pupil diameter for each paired recording (magenta dot: mean). Both distributions are significantly larger than 0 (peak correlation coefficient: 0.15 ± 0.02, p=8.3e-7, signed rank = 743; time lags: 2.61 ± 0.39 s, p=7.8e-7, signed rank = 744, n = 39). (**c**) Grand average trial-aligned LC spike rate (n = 43, top), and pupil diameter (n = 36, bottom) averaged by different trial types. Gray and black arrows indicate tone and stimulus onsets, respectively. In Hit trials, the latency of LC responses to tone onset was 0.064 ± 0.005 s, and to whisker stimulation onset was 0.111 ± 0.008 s, and the reaction time (first lick latency to whisker stimulation onset) of the mice was 0.58 ± 0.03 s. (**d**) Left: LC responses to tone (T) and Go responses (G) during Hit trials with median indicated. Tone vs. Go: 4.79 (3.70–6.66) sp/s vs. 4.68 (3.33–7.26) sp/s, median (IQR), p=0.24, signed rank = 496.5, n = 43. Right: Pupil responses to tone and Go responses during Hit trials with median indicated. Tone vs. Go: 0.003 (−0.015–0.015) mm vs. 0.027 (−0.010–0.063) mm, median (IQR), p=6.4e-5, signed rank = 559, n = 36. Gray lines indicate individual recordings. (**e**) Top: Pre-stimulus (baseline) and post-stimulus (evoked) LC spike rate for Hit and Miss trials with median indicated (Baseline: Hit vs. Miss, 0.66 (0.30–3.51) sp/s vs. 1.55 (0.68–3.00) sp/s, median (IQR), p=0.0083, signed rank = 254.5; Evoked: Hit vs. Miss, 3.24 (1.78–5.49) sp/s vs. 1.82 (0.95–3.45) sp/s, median (IQR), p=5.5e-7, signed rank = 782.5, n = 43). Gray lines indicate individual recordings. Bottom: Histogram of choice probability for Hit vs. Miss trials based on baseline and evoked LC activity (magenta dots: mean). Choice probabilities are significantly deviated from 0.5. Baseline: 0.47 ± 0.014, p=0.032, signed rank = 295.5; Evoked: 0.59 ± 0.017, p=4.6e-6, signed rank = 751, n = 43. (**f**) Lick-aligned LC spike rate (top) and pupil diameter (ΔPupil, bottom) averaged by trial types: Hit (blue), FA (green).

The online version of this article includes the following source data and figure supplement(s) for figure 2:

**Source data 1.** MATLAB R2016b file with data shown in panels b, d and e.
**Figure supplement 1.** LC responses during Hit and Miss trials.
**Figure supplement 1—source data 1.** MATLAB R2016b file with data shown in panels a and b.
**Figure supplement 2.** Pupil responses to the tone during Hit and Miss trials.
**Figure supplement 2—source data 1.** MATLAB R2016b file with pupil and choice probability data shown in panels a and b.

pupil dilation following LC spikes (peak correlation coefficient: 0.15 ± 0.02; time lags: 2.61 ± 0.39 s, n = 39 recordings, *Figure 2b*). Mean LC spiking activity aligned with trial onsets showed prominent responses to a tone delivered at the beginning of each trial, as well as in trials where mice made Go (licking) responses (Hit and False Alarm trials, *Figure 2a,c*). LC spiking activity to the tone was comparable to Go responses (p=0.24, *Figure 2d*, Materials and methods). On Hit trials, where mice successfully licked to the whisker stimulus, pre-stimulus LC activity (measured in a 0.5 s window prior to stimulus onset) was slightly but significantly lower than Miss trials, where mice failed to lick to the whisker stimulus (*Figure 2e*). We note that on Miss trials, LC responded weakly to the whisker stimulus alone (<0.5 sp/s above baseline, *Figure 2—figure supplement 1*). LC activity measured in a short window (0.2 s) after stimulus onset was larger on Hits compared with Misses (*Figure 2e*; the same trend holds for 0.1 s window, data not shown). Ideal-observer analysis showed that both pre- and post-stimulus LC activity significantly predicted perceptual reports of the mice on a trial-by-trial basis, with choice probabilities (*Yang et al., 2016*) of 0.47 ± 0.014 (p=0.032, n = 43) for pre-stimulus and 0.59 ± 0.017 (p=4.6e-6, n = 43) for post-stimulus LC activity, respectively (*Figure 2e*). LC activity aligned to the time of licking showed that spiking responses began ~200 ms prior to licking (*Figure 2f*).

In striking contrast, pupil diameter minimally increased in response to the tone. Instead, pupil strongly dilated on Hit and False Alarm trials, in which mice made Go (licking) responses (*Figure 2a, c,d*; tone vs. Go: p=6.4e-5, n = 36, Materials and methods) (*Lee and Margolis, 2016*). Interestingly, pupil response to the tone was larger on Misses compared to Hits and significantly predicted perceptual choices of the mice (*Figure 2—figure supplement 2*). Pupil diameter changes (ΔPupil) aligned to the time of licking showed that pupil responses occurred after licking (*Figure 2f*).

Together, these data show that LC and pupil responses were positively correlated. Both LC activity and pupil diameter increased during licking responses, but LC also strongly responded to the tone, a salient sensory cue that alerted mice to trial onsets. Thus, LC activity and pupil diameter appear to reflect different sets of task events during this behavior.

Next, we analyzed recordings where we simultaneously measured membrane potential ($V_m$) of S1 neurons (mostly from layer 2/3, *Figure 3—figure supplement 1*), along with LC spiking and/or pupil diameter during the detection task. Our goal was to determine how LC spiking related to cortical activity and to pupil diameter during task performance. We used spike-triggered averages (STAs) to quantify how individual spikes from single LC units correlated with changes in $V_m$ and pupil

diameter. LC spike-triggered $V_m$ analyses revealed that LC spikes were associated with a depolarization in cortical neurons (1.39 ± 0.35 mV, n = 12, *Figure 3a–c*). On average, $V_m$ depolarization associated with an LC spike peaked after the spike, with short time lags from an LC spike to peak depolarization in S1 (0.17 ± 0.06 s, n = 12, *Figure 3a–c*; also see *Figure 3—figure supplement 2*).

Consistent with the previous cross-correlogram analysis based on a larger set of LC–pupil recordings (*Figure 2b*), here STA analysis showed that pupil diameter increased in association with individual spikes from LC single units (0.03 ± 0.01 mm, n = 7), with peak dilation occurring roughly 10-fold slower than peak $V_m$ depolarization (time lags from an LC spike to peak pupil dilation: 1.89 ± 0.25 s, n = 7, *Figure 3d–f*).

Given that pupil diameter and LC activity are positively correlated, and that pupil diameter has been often considered to index LC activity (*Aston-Jones and Cohen, 2005*; *McGinley et al., 2015b*), we next tested whether the pupil–S1 relationship resembled the LC–S1 relationship. Cross-correlogram analyses revealed heterogeneous correlations between pupil diameter and S1 $V_m$, with both positive and negative correlations as well as positive and negative time lags (peak correlation coefficient: 0.05 ± 0.04; time lags: −0.22 ± 1.01 s, n = 19, *Figure 3g–i*). In sharp contrast, the time derivative of pupil diameter (pupil') was positively correlated with S1 $V_m$ in a more consistent manner (peak correlation coefficient: 0.15 ± 0.03; time lags: 1.31 ± 0.24 s, n = 19, *Figure 3j,k*; *Reimer et al., 2016*; *Reimer et al., 2014*). We further examined how well LC spiking and pupil diameter can predict cortical $V_m$ fluctuations at different timescales. We found that LC activity was superior in predicting cortical dynamics faster than ~200–300 ms (exponential decay time constant: LC–$V_m$ vs. Pupil–$V_m$ vs. Pupil'-$V_m$, 1.02 ± 0.09 vs. 6.59 ± 0.60 vs. 1.53 ± 0.14, *Figure 3l*; repeated-measures ANOVA, F(2, 36)=74.5, p=1.6e-13, n = 19). Post hoc Tukey–Kramer tests revealed that LC activity reflects higher frequency correlations with $V_m$ compared with the pupil (LC–$V_m$ vs. Pupil–$V_m$, p=5.9e-8; LC-$V_m$ vs. Pupil'–$V_m$, p=0.0037; Pupil–$V_m$ vs. Pupil'–$V_m$, p=7.1e-7). On the other hand, the LC–Pupil' relationship was similar to that of LC–Pupil (compare *Figure 3—figure supplement 3* with *Figure 3e,f*).

Together, these data show that LC spikes preceded S1 depolarizations and pupil dilations. LC spiking correlated with both $V_m$ and pupil diameter changes, but on vastly different timescales (~0.2 s vs. ~2 s). Our data also show that the time derivative of pupil diameter, but not the absolute pupil size, is a good predictor of S1 $V_m$ fluctuations. However, LC spiking can track fast $V_m$ fluctuations better than either pupil or pupil'.

Individual trials in our detection task contained distinct events, including the tone that alerted mice of the trial start ('Tone'), the whisker stimulus on Go trials ('Stimulus'), and licks ('Lick'), as well as other periods in which mice did not receive stimuli or make lick responses ('Quiet'). For a more granular perspective on how LC spiking correlated with changes in $V_m$ and pupil diameter, we computed LC spike-triggered averages separately in these different event windows (task epochs, Materials and methods).

While single LC spikes were associated with prominent changes in both cortical $V_m$ and pupil diameter, we found that these associations strikingly depended on task epoch: $V_m$ depolarization associated with an LC spike had the biggest response to tone/licking and almost no response during the quiet periods (*Figure 4a*). In contrast, pupil dilation associated with an LC spike had the biggest response to licking and almost no response to the tone (*Figure 4b*; this was true as well for pupil dilation after z-scoring, *Figure 4—figure supplement 1*). The pupil' associated with an LC spike had an intermediate response to the tone (*Figure 4c*). In addition, peak pupil dilation, pupil' and $V_m$ depolarization appeared to have different dependencies on LC spike counts, with a roughly monotonic relationship between pupil and LC, and a much weaker dependence between $V_m$ and LC (*Figure 4—figure supplement 2*). The relationship between $V_m$ and LC spike count could partly but not fully be attributable to differences in the LC inter-spike intervals (*Figure 4—figure supplement 2a*). Thus, the correlations between LC spiking and $V_m$, and between LC spiking and pupil diameter, are non-stationary, even on the timescale of a few seconds. Importantly, these epoch dependencies were different for $V_m$ and pupil – with the biggest response occurring to the tone for $V_m$ and the smallest response occurring to the tone for pupil – suggesting that the correlations between LC activity and $V_m$ and pupil each reflect distinct unmeasured underlying processes.

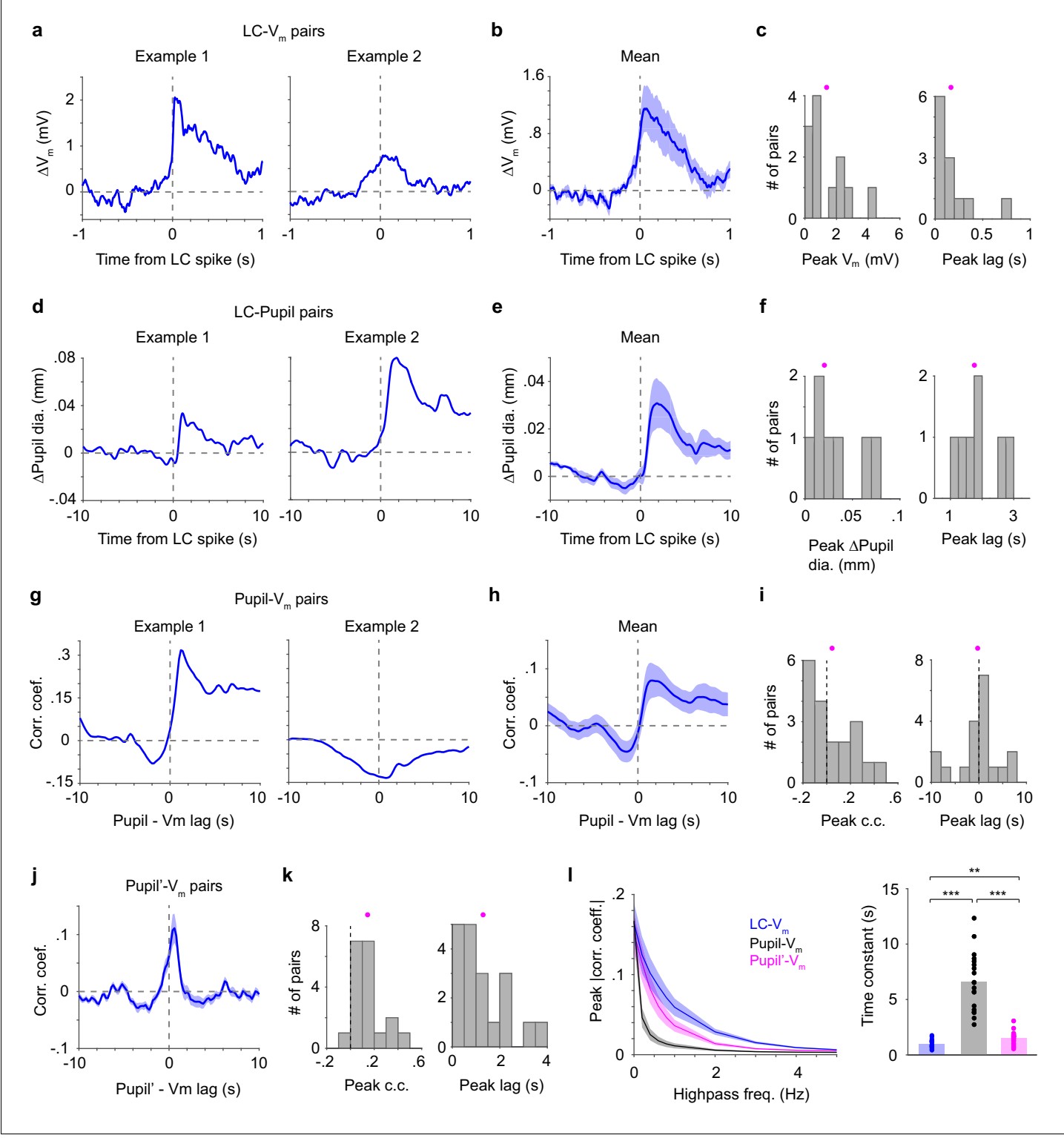

**Figure 3.** Different relationships between LC spikes, S1 $V_m$, and pupil diameter. (a) Two examples of LC spike-triggered average $\Delta V_m$. (b) Group mean of LC spike-triggered average $\Delta V_m$ (± s.e.m., n = 12). (c) Histograms of peak $\Delta V_m$ and peak lags (showing all LC–S1 pairs) with means indicated (magenta dots). Both distributions are significantly larger than 0 (peak $\Delta V_m$: 1.39 ± 0.35 mV, p=4.9e-4, signed rank = 78; peak lags: 0.17 ± 0.06 s, p=4.9e-4, signed rank = 78, n = 12). (d) Two examples of LC spike-triggered average $\Delta$Pupil. (e) LC spike-triggered average $\Delta$Pupil group mean (± s.e.m., n = 7). (f) Histograms of peak $\Delta$Pupil and peak lags (showing all LC–Pupil pairs) with means indicated (magenta dots). Both distributions are significantly larger than 0 (peak $\Delta$Pupil: 0.03 ± 0.01 mm, p=0.016, signed rank = 28; peak lags: 1.89 ± 0.25 s, p=0.016, signed rank = 28, n = 7). (g) Two examples of

*Figure 3 continued on next page*

*Figure 3 continued*

Pupil–$V_m$ cross-correlograms. (**h**) Group mean of Pupil–$V_m$ cross-correlograms (± s.e.m., n = 19). (**i**) Histograms of peak Pupil–$V_m$ correlation coefficient and peak lags (showing all S1–Pupil pairs) with means indicated (magenta dots). Both distributions are not significantly deviated from 0 (peak correlation coefficient: 0.05 ± 0.04, p=0.33, signed rank = 119; peak lags: - 0.22 ± 1.01 s, p=0.87, signed rank = 99, n = 19). (**j**) Group mean of the time derivative of pupil (Pupil')–$V_m$ cross-correlograms (± s.e.m., n = 19). (**k**) Histograms of peak Pupil'–$V_m$ correlation coefficient and peak lags with means indicated (magenta dots). Both distributions are significantly larger than 0 (peak correlation coefficient: 0.15 ± 0.03, p=1.6e-4, signed rank = 189; peak lags: 1.31 ± 0.24 s, p=1.3e-4, signed rank = 190, n = 19). (**l**) Left: Peak correlation coefficient for LC–$V_m$, Pupil–$V_m$ and Pupil'–$V_m$ pairs after progressive high-pass filtering of S1 $V_m$. Right: Exponential decay functions (corr. coef. = a*exp(−freq*μ)) were fitted to these curves. The time constant μ is significantly different (repeated-measures ANOVA, F(2, 36)=74.5, p=1.6e-13, n = 19). Post hoc Tukey–Kramer tests revealed that the LC–$V_m$ relationship had the slowest decay and Pupil–$V_m$ had the fastest decay. LC–$V_m$ vs. Pupil–$V_m$, p=5.9e-8; LC–$V_m$ vs. Pupil'–$V_m$, p=0.0037; Pupil–$V_m$ vs. Pupil'–$V_m$, p=7.1e-7.

The online version of this article includes the following source data and figure supplement(s) for figure 3:

**Source data 1.** MATLAB R2016b file with data shown in *Figure 3* c, f, i, k and l.
**Figure supplement 1.** Histograms of the depth of S1 whole-cell recordings.
**Figure supplement 1—source data 1.** MATLAB R2016b file with recording depth data.
**Figure supplement 2.** Cross-correlation between LC spikes and S1 $V_m$.
**Figure supplement 2—source data 1.** MATLAB R2016b file with data shown in panel b.
**Figure supplement 3.** LC spike-triggered time derivative of pupil diameter.
**Figure supplement 3—source data 1.** MATLAB R2016b file with data shown in panel b.

## Discussion

Tonic LC activity is thought to reflect attentive state (*Usher et al., 1999*; *Aston-Jones and Cohen, 2005*). We found that pre-stimulus baseline LC spiking predicted behavioral responses, suggesting that being in a hyperactive or distractable state underlies failed stimulus detection. We excluded trials where mice licked before stimulus onset from our analysis; thus, the higher baseline spiking in Miss trials was not caused by premature licking. However, it is possible that higher baseline LC activity was correlated with more whisking, which has been shown to lead to failed stimulus detection (*Ollerenshaw et al., 2012*; *Kyriakatos et al., 2017*). Overall, the effect was weak, possibly due to the use of an auditory cue that puts the mice in a more homogeneous arousal/attentive state. In other tasks without such alerting cues, task performance may have a stronger dependence on arousal and pre-stimulus LC activity.

Recent optogenetic and pharmacological work has shown that activating the LC–NE system enhances detection/discrimination sensitivity (*Gelbard-Sagiv et al., 2018*; *Rodenkirch et al., 2019*; *McBurney-Lin et al., 2020*). Thus, LC–NE acting on early sensory areas (i.e., primary sensory thalamus, primary sensory cortex) likely affects perceptual decision making (*McBurney-Lin et al., 2019*). We showed that S1 depolarizations immediately follow LC spiking, suggesting that LC activity at least partially contributes to the amount of depolarization in S1 neurons (*Constantinople and Bruno, 2011*; *Polack et al., 2013*). Given that sensory-evoked S1 responses predict the behavioral choices of the mice (*Sachidhanandam et al., 2013*; *Yang et al., 2016*; *Kwon et al., 2016*), our results suggest that the LC–S1 pathway is involved during this task. However, the effect could be mediated directly through LC innervation of S1 or indirectly through LC innervating other early somatosensory areas, such as the VPM thalamus (*Rodenkirch et al., 2019*). Future gain- and loss-of-function experiments are needed to determine causal roles of these distinct pathways to S1 and to determine how different LC activity patterns (e.g., tonic vs. phasic) affect sensory processing and task performance.

LC responded strongly to an auditory cue (tone) meant to alert the mice to the beginning of a trial. Such responses may be mediated via synaptic inputs to LC from the brainstem (e.g., nucleus paragigantocellularis) (*Ennis and Aston-Jones, 1988*; *Aston-Jones et al., 1991*; *Van Bockstaele et al., 1993*; *Llorca-Torralba et al., 2016*). While this tone carried no information about the presence of a tactile stimulus or reward on any given trial, and therefore was not associated with a particular movement response, it did inform the mice about the time when a tactile stimulus could occur (in our task the duration between the tone and stimulus onset was fixed). The robust LC spiking responses to this cue are therefore consistent with LC's role in promoting alertness or preparedness to detect a weak stimulus (*Berridge and Waterhouse, 2003*; *Aston-Jones and Cohen, 2005*). We also found that LC responded to operant licking responses, which is consistent with earlier work showing that LC encoded overt decision execution (*Kalwani et al., 2014*).

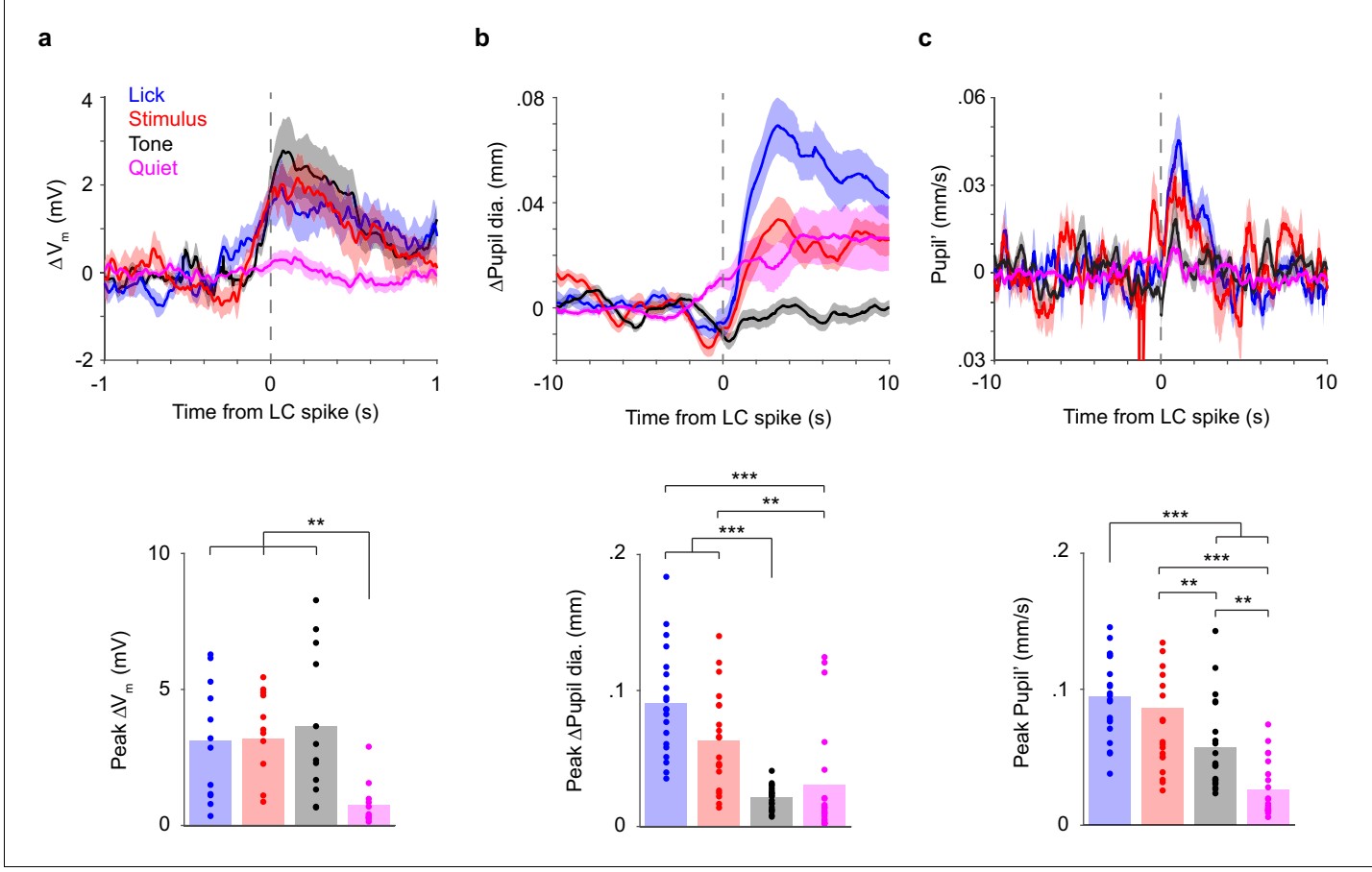

**Figure 4.** Correlations between LC spikes, S1 $V_m$, and pupil diameter depend on task epoch. (a) Top: LC spike-triggered $\Delta V_m$ separated by task epoch: tone, stimulus, lick, and quiet. Bottom: Bar graphs of peak $\Delta V_m$ for each epoch. Dots indicate individual paired recordings. Repeated-measure ANOVA, $F_{(3, 33)}$=9.2, p=1.4e-4, n = 12. Post hoc Tukey–Kramer tests revealed that peak $\Delta V_m$ in lick, stimulus, and tone epochs were not different from each other. Lick vs. Stim, p=1.00; Lick vs. Tone, p=0.76; Stim vs. Tone, p=0.94. Peak $\Delta V_m$ in quiet epochs was lower. Quiet vs. Lick, p=0.0059; Quiet vs. Stim, p=0.0038; Quiet vs. Tone, p=0.0041. (b) Top: LC spike-triggered $\Delta$Pupil separated by task epoch. Bottom: Bar graphs of peak $\Delta$Pupil for each epoch. Dots indicate individual paired recordings. Repeated-measure ANOVA, $F_{(3, 57)}$=22.1, p=1.3e-9, n = 20. Post hoc Tukey–Kramer tests revealed that peak $\Delta$Pupil in lick and stimulus epochs were larger than in tone and quiet epochs. Lick vs. Stim, p=0.10; Tone vs. Quiet, p=0.76; Lick vs. Tone, p=3.7e-7; Lick vs. Quiet, p=6.2e-4; Stim vs. Tone, p=1.1e-4; Stim vs. Quiet, p=0.0027. (c) Top: LC spike-triggered pupil' separated by task epoch. Bottom: Bar graphs of peak pupil' for each epoch. Dots indicate individual paired recordings. Repeated-measures ANOVA, $F_{(3, 57)}$=35.3, p=4.9e-13, n = 20. Post hoc Tukey–Kramer tests revealed that peak pupil' in lick and stimulus epochs were larger than in tone, and peak pupil' in quiet epochs was the lowest. Lick vs. Stim, p=0.46; Tone vs. Quiet, p=0.0013; Lick vs. Tone, p=1.0e-4; Lick vs. Quiet, p=1.4e-8; Stim vs. Tone, p=0.0058; Stim vs. Quiet, p=6.7e-6.

The online version of this article includes the following source data and figure supplement(s) for figure 4:

**Source data 1.** MATLAB R2016b file with data shown in *Figure 4a–c*.

**Figure supplement 1.** Correlations between LC spikes and z-scored pupil diameter depend on task epoch.

**Figure supplement 1—source data 1.** MATLAB R2016b file with data shown in panel b .

**Figure supplement 2.** Dependencies of pupil diameter, pupil' and $V_m$ depolarization on LC spike count.

**Figure supplement 3.** Analysis using an alternative window to define the 'Tone' epoch.

**Figure supplement 3—source data 1.** MATLAB R2016b file with data shown in panels a,b.

Our data show that while LC spiking and pupil diameter correlate well at long timescales, and both can predict changes in cortical dynamics, LC does so an order of magnitude faster. Moreover, the correlation between pupil and $V_m$ is much more heterogeneous than between LC and $V_m$. In support of previous studies, our results suggest that compared with change in the absolute size of pupil diameter, its time derivative is a better predictor of cortical states (*Reimer et al., 2014*; *Reimer et al., 2016*). Importantly, the relationships between LC activity, S1 $V_m$, and pupil depended on task epoch. Because these epochs changed on the timescale of a few seconds, our data imply

that pupil diameter can be dissociated from LC spiking and cannot be used as a stationary index of LC activity. However, comparing across repeats of similar epochs should yield a more accurate prediction of LC spiking by pupil diameter. That is, in attempting to use pupil diameter as a proxy for LC spiking, our data suggest it would be useful to separately normalize distinct task epochs. Future work should examine the LC–pupil relationship using fine-scale analyses that consider behavioral states at a granular level specific to individual tasks.

Pupil size changes have been linked to activity in multiple brain areas and neuromodulatory systems (*Joshi et al., 2016*; *Reimer et al., 2016*; *Cazettes et al., 2021*), and different pupil response profiles reflect different cognitive processes (*Schriver et al., 2020*). Therefore, it is possible that the pupil exhibits dynamic coupling with different underlying brain circuits in a cognitive process (behavioral epoch)-dependent way. In addition, a recent study showed that pupil responses to dorsal raphe stimulation exhibited task uncertainty-dependent variations (*Cazettes et al., 2021*). Thus, it is also possible that other modulatory systems (e.g., serotonergic and cholinergic, *Berridge and Waterhouse, 2003*) modulate the pupil–LC coupling in a dynamic manner.

# Materials and methods

## Key resources table

| Reagent type (species) or resource | Designation | Source or reference | Identifiers | Additional information |
|---|---|---|---|---|
| Strain, strain background (*M. musculus*) | DBH-Cre | MMRRC | Cat# 036778-UCD, RRID:MMRRC_036778-UCD | |
| Strain, strain background (*M. musculus*) | Ai32 | Jackson Laboratory | Cat#: JAX:012569, RRID:IMSR_JAX:012569 | |
| Software, algorithm | BControl | Princeton University | https://brodylabwiki.princeton.edu/bcontrol | |
| Software, algorithm | WaveSurfer | HHMI Janelia | http://wavesurfer.janelia.org/ | |
| Software, algorithm | MATLAB | MathWorks | RRID:SCR_001622 | |
| Software, algorithm | StreamPix | Norpix | RRID:SCR_015773 | |
| Software, algorithm | Adobe Illustrator | Adobe | RRID:SCR_010279 | |
| Other | High-speed CMOS camera | PhotonFocus | DR1-D1312-200-G2-8 | |
| Other | Telecentric lens | Edmund Optics | Cat#: 55–349 | |
| Other | Pipette glass | Warner Instruments | Cat#: 640792 | |
| Other | Tetrode drive | *Cohen et al., 2012* | N/A | |
| Antibody | Anti-TH primary antibody (rabbit, polyclonal) | Thermo-Fisher | Cat#: OPA1-04050, RRID: AB_325653 | (1:1000) |
| Antibody | Secondary antibody (goat, polyclonal) | Thermo-Fisher | Cat#: A-11008, RRID:AB_2534079 | (1:500) |

All procedures were performed in accordance with protocols approved by the Johns Hopkins University Animal Care and Use Committee. Mice were DBH-Cre (B6.FVB(Cg)-Tg(Dbh-cre) KH212Gsat/Mmucd, 036778-UCD, MMRRC) and Ai32 (RCL-ChR2(H134R)/EYFP, 012569, JAX), singly housed in a vivarium with reverse light–dark cycle (12 hr each phase). Male and female mice of 6–12 weeks were implanted with titanium head posts as described previously (*Yang et al., 2016*). After recovery, mice were trained to perform a Go/NoGo single whisker detection task as described previously (*Yang et al., 2016*). Behavioral apparatus was controlled by BControl (C. Brody, Princeton University). A custom 'lickport' was placed within reach of the mouse's tongue and used both to deliver water rewards and to record the time of each lick, determined via measurement of an electrical

conductance change caused by contact between tongue and lickport. On go trials, a single whisker was deflected for 0.5 s with a 40 Hz sinusoidal deflection (rostral to caudal, peak angular speed ~800 deg/s). A 'response window' was defined as 0.2–2 s after the time of whisker stimulus onset for go trials or the time that the whisker stimulus would have onset on no-go trials. A 'hit' trial occurred when mice licked the lickport within the response window, and a drop of water was delivered. On go trials, if mice did not lick within the 1.8 s response window, the trial was scored as a 'miss' and no reward or punishment was delivered. Go trials were randomly mixed with no-go trials, in which the whisker was not deflected. No more than three consecutive trials of the same type were allowed. On no-go trials, if mice licked within the response window, it was scored as a 'false alarm', and mice were punished with a 3–5 s time-out. If mice licked during the time-out, an additional time-out was triggered. A 'correct rejection' occurred when mice withheld licking during the response window. Correct rejections were not rewarded. A 0.1 s auditory cue (8 kHz tone, ~80 dB SPL) was introduced starting 1 s before stimulus onset. During all sessions, ambient white noise (cut off at 40 kHz, ~80 dB SPL) was played through a separate speaker to mask any other potential auditory cues associated with movement of the piezo stimulator. Trials where mice made 'premature' licking during the period between tone and 0.1 s after whisker stimulation onset were excluded from further analysis. To align LC activity and pupil traces to the 'first lick' (*Figure 2f*), we used the first lick occurring in the response window. A total of 46 recordings from eight mice are reported (mean hit rate: 0.52 ± 0.03; false alarm rate: 0.11 ± 0.02).

Custom microdrives with eight tetrodes and an optic fiber (*Cohen et al., 2012*, 0.39 NA, 200 μm core) were built to make extracellular recordings from LC neurons. Each tetrode comprised four nichrome wires (100–300 kΩ). A ~1 mm diameter craniotomy was made (centered at −5.2 mm caudal and 0.85 mm lateral relative to bregma) for implanting the tetrodes to a depth of 2.7 mm relative to the brain surface. The microdrive was advanced in steps of ~100 μm each day until reaching LC, identified by optogenetic tagging of DBH+ neurons expressing ChR2, tail pinch response, wide extracellular spike waveforms, and post hoc electrolytic lesions. Broadband voltage traces were acquired at 30 kHz (Intan Technologies) and filtered between 0.1 and 10 kHz. Signals were then bandpass filtered between 300 and 6000 Hz, and spikes were detected using a threshold of 4–6 standard deviations. The timestamp of the peak of each detected spike, as well as a 1 ms waveform centered at the peak, was extracted from each channel for offline spike sorting using MClust (*Redish, 2014*). At the conclusion of the experiments, brains were perfused with PBS followed by 4% PFA, post-fixed overnight, then cut into 100 μm coronal sections, and stained with anti-tyrosine hydroxylase antibody (Thermo-Fisher OPA1-04050).

Pupil video was acquired at 50 Hz using a PhotonFocus camera and StreamPix five software. Light from a 940 nm LED was passed through a condenser lens and directed to the right eye, reflected off a mirror, and directed into a 0.25× telecentric lens. WaveSurfer (https://www.janelia.org/open-science/wavesurfer) triggered individual camera frames synchronized with electrophysiological recordings.

In a subset of animals, we performed simultaneous intracellular current clamp (whole-cell) recordings in conjunction with LC recording and/or pupil tracking during behavior. A craniotomy over the C2 barrel was made based on intrinsic signal imaging (*Yang et al., 2016*). In some cases, we also made craniotomies over nearby barrels based on the known somatotopy of S1 (*Welker and Woolsey, 1974*; *Wilson et al., 2000*) to increase yield. Whole-cell recording procedures, quality control, and data processing were performed as described previously (*Yang et al., 2016*). Briefly, borosilicate glass pipettes (1.5 mm OD, 0.86 mm ID; Harvard Apparatus) were pulled (P-97, Sutter) to have a long shank and were 4–7 MΩ when filled with solution containing (in mM): 135 potassium gluconate, 4 KCl, 10 sodium phos-phocreatine, 4 ATP magnesium salt, 0.3 GTP sodium salt hydrate, 10 HEPES, 3 mg/ml biocytin (pH 7.3 with KOH). Electrophysiological signals (Multiclamp 700B, Molecular Devices) were filtered at 10 kHz and acquired at 20 kHz using Ephus or WaveSurfer.

For *Figure 2d*, LC responses to the tone were calculated using a 300 ms window starting at tone onset, and LC responses to Go were calculated using a 300 ms window starting 200 ms after stimulus onset to capture peak responses. These estimates were based on LC response profiles in *Figure 2c*. Pupil responses to the tone were calculated using a 1 s window starting 1 s after tone onset. This estimate was primarily based on the pupil response profile during CR trials (e.g., *Figure 2a,c*, indicated by th gray bar), where there was no whisker stimulus or licking response. Pupil responses to Go (licking) were calculated using a 1 s window starting 1.5 s after stimulus onset (e.g.,

False Alarm trials in *Figure 2a,c*, indicated by the black bar). Based on the temporal profiles of pupil diameter in different trial types shown in *Figure 2a,c*, and that the whisker stimulus started 1 s after tone onset, pupil responses to tone and Go can be segregated. These estimates were consistent with the results showing that pupil dilated 1–2 s after LC spikes (*Figures 2b* and *3d-f*).

For *Figure 2e*, pre-stimulus LC baseline activity was calculated using a 500 ms window ending 50 ms before stimulus onset. Post-stimulus activity was calculated using a 200 ms window starting 20 ms after stimulus onset, before licking responses (*Yang et al., 2016*). Choice probabilities were computed as described previously (*Yang et al., 2016*).

To compute lick-aligned changes in LC spiking and pupil diameter, we only used licks that occurred at least 0.5 s after the previous lick. To compute LC spike-triggered S1 $V_m$ and pupil, we only used LC spikes that occurred at least 0.5 s after the previous spike. For STA analysis, peak $\Delta V_m$, $\Delta$Pupil, or the time derivative of Pupil (Pupil') were defined as the largest positive or negative value within the observed window ($\pm 1$ s or $\pm 10$ s, respectively).

For cross-correlogram analysis, each LC spike train was convolved with a 400 ms wide Gaussian kernel (results hold for 200 ms kernel, data not shown). Peak correlation coefficients were defined as the largest positive or negative value within the observed window ($\pm 1$ s or $\pm 10$ s). To examine how well LC spiking and pupil diameter could predict cortical $V_m$ fluctuations at different timescales (*Figure 3l*), $V_m$ was high-pass filtered at 0, 0.2, 0.4, 0.6, 0.8, 1, 2, 3, 4, and 5 Hz separately. Cross-correlogram analysis between the filtered $V_m$ and LC (pupil) activities was then performed as described above, and largest absolute values of peak correlation coefficients were taken.

Task epochs were defined as follows: 'Tone' epochs: $-0.2$ s to 0.3 s with respect to tone onset (we also tested 0 to 0.5 s, *Figure 4—figure supplement 3*); 'Stimulus' epochs: $-0.2$ s to 0.3 s with respect to stimulus onset (i.e., only on trials with whisker stimulation); 'Licking' epochs: $-0.2$ s to 0.3 s with respect to licks that occurred at least 0.5 s after the previous lick; 'Quiet' epochs: non-overlapping 0.5 s segments excluding the three types of epoch defined previously during the entire session.

Thirty-nine LC–pupil pairs from seven mice were included in *Figure 2b*, including single- and multi-units, with and without S1 recordings. For the rest of *Figure 2*, LC analysis included 43 recordings from eight mice, each with at least four Hit and four Miss trials. Among those, 36 were with pupil recordings and were used for pupil analysis. Twelve pairs of S1 whole-cell and LC single-unit recordings from six mice were included in *Figures 3a–c* and *4a*, of which seven were with pupil recordings and included in *Figure 3d–f*. Nineteen S1–pupil recordings from three mice were included in *Figure 3g–l*. Twenty pairs of LC SU and pupil recordings from five mice were included in *Figure 4b,c*, with and without S1 recordings.

Data were reported as mean ± s.e.m. unless otherwise noted. Statistical tests were by two-tailed Wilcoxon signed-rank unless otherwise noted. We did not use statistical methods to predetermine sample sizes. Sample sizes are similar to those reported in the field. We assigned mice to experimental groups arbitrarily, without randomization or blinding.

## Acknowledgements

We thank Edward Zagha for comments on the manuscript and Dwight E Bergles for DBH-Cre mice. This work was supported by UCR startup (HY), Klingenstein-Simons Fellowship Awards in Neuroscience (HY), NIH grants F30MH110084 (BAB), 1R01NS107355 (HY), 1R01NS112200 (HY), R01NS089652 (DHO), 1R01NS104834-01 (DHO, JYC).

## Additional information

### Funding

| Funder | Grant reference number | Author |
| --- | --- | --- |
| National Institute of Neurological Disorders and Stroke | R01NS089652 | Daniel H O'Connor |
| National Institute of Neurological Disorders and Stroke | 1R01NS104834-01 | Jeremiah Y Cohen<br>Daniel H O'Connor |
| National Institute of Neurological Disorders and Stroke | 1R01NS107355 | Hongdian Yang |

| National Institute of Neurological Disorders and Stroke | 1R01NS112200 | Hongdian Yang |
| National Institute of Mental Health | F30MH110084 | Bilal A Bari |
| Klingenstein-Simons | Fellowship Awards in Neuroscience | Hongdian Yang |

The funders had no role in study design, data collection and interpretation, or the decision to submit the work for publication.

## Author contributions

Hongdian Yang, Conceptualization, Data curation, Software, Formal analysis, Validation, Investigation, Visualization, Methodology, Writing - original draft, Writing - review and editing; Bilal A Bari, Formal analysis, Methodology; Jeremiah Y Cohen, Conceptualization, Resources, Funding acquisition, Methodology; Daniel H O'Connor, Conceptualization, Resources, Data curation, Software, Formal analysis, Supervision, Funding acquisition, Validation, Visualization, Methodology, Writing - original draft, Project administration, Writing - review and editing

## Author ORCIDs

Hongdian Yang https://orcid.org/0000-0002-5203-9519
Jeremiah Y Cohen http://orcid.org/0000-0002-4768-7124
Daniel H O'Connor https://orcid.org/0000-0002-9193-6714

## Ethics

Animal experimentation: All procedures were performed in accordance with protocols (#MO18M187) approved by the Johns Hopkins University Animal Care and Use Committee.

## Decision letter and Author response

Decision letter https://doi.org/10.7554/eLife.64327.sa1
Author response https://doi.org/10.7554/eLife.64327.sa2

# Additional files

## Supplementary files

• Transparent reporting form

## Data availability

Source data for main figures (Figures 2–4) and figure supplements (Figure 1-figure supplement 1; Figure 2-figure supplement 1; Figure2-figure supplement 2; Figure3-figure supplement 1; Figure3-figure supplement 2; Figure3-figure supplement 3; Figure4-figure supplement 1; Figure4-figure supplement 2; Figure4-figure supplement 3) are uploaded as 'Source data' files.

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
