## [Decision Letter]

**Acceptance summary:**

This paper will be of interest to neuroscientists studying the effect of brain state on sensory processing and behavior and the effects of noradrenergic input from the locus coeruleus on cortex. By combining multiple types of recordings in the same behaving animal, the authors are able to build on and extend previous work that has demonstrated some of these effects in separate experiments.

**Decision letter after peer review:**

Thank you for submitting your article "Locus coeruleus spiking differently correlates with somatosensory cortex activity and pupil diameter" for consideration by *eLife*. Your article has been reviewed by three peer reviewers, one of whom is a member of our Board of Reviewing Editors, and the evaluation has been overseen by Joshua Gold as the Senior Editor. The following individuals involved in review of your submission have agreed to reveal their identity: Qi Wang (Reviewer #2); Carl C H Petersen (Reviewer #3).

The reviewers have discussed the reviews with one another and the Reviewing Editor has drafted this decision to help you prepare a revised submission.

As the editors have judged that your manuscript is of significant interest, but as described below that additional analyses are required before it is published, we would like to draw your attention to changes in our revision policy that we have made in response to COVID-19 (https://elifesciences.org/articles/57162). First, because many researchers have temporarily lost access to the labs, we will give authors as much time as they need to submit revised manuscripts. We are also offering, if you choose, to post the manuscript to bioRxiv (if it is not already there) along with this decision letter and a formal designation that the manuscript is "in revision at *eLife*". Please let us know if you would like to pursue this option. (If your work is more suitable for medRxiv, you will need to post the preprint yourself, as the mechanisms for us to do so are still in development.)

Summary:

Yang et al., make careful measurements of the spiking activity of optogenetically-identified locus coeruleus (LC) neurons during a whisker detection task in head-restrained mice together with measurement of neuronal membrane potential in the primary somatosensory cortex (S1) and pupil diameter. A substantial body of previous work has linked LC activity with cortical intracellular membrane potential (V_m_) depolarization, pupil dilation, and behavioral performance, but simultaneous measurements of all of three of these correlates in the same behaving animal – as the authors perform here – are rare and technically challenging, and these experiments offer the potential to study features of timing and trial-to-trial variability that might otherwise be inaccessible. The authors focus on two issues in particular: First, the relative ability of pupil dilation and "ground truth" recordings of LC firing to predict behavioral performance and membrane depolarization in sensory cortex. Perhaps not surprisingly, both pupil and LC are informative, but the time scale of pupil correlations with behavior and intracellular membrane potentials is an order of magnitude longer (1s^-1^0s of seconds), compared to LC firing. Second, the authors discovered that the impact of LC spikes upon S1 membrane potential and pupil diameter differed across behavioural epochs. Given that pupil diameter is often interpreted as a direct correlate of LC activity, the results may be of broad interest. There are many potential factors that likely contribute to this heterogeneity across different task epochs that the authors do not explore, including other neuromodulator systems like acetylcholine, and population-level changes in cortical activity, but reviewers agreed that the high quality of the experimental methodology make the data strong.

Essential revisions:

1) Given the qualitative difference between the tonic and phasic modes in the LC, it would be helpful to have a plot like Figure 4—figure supplement 2, but conditioned on different inter-spike intervals. Specifically it would be interesting to know whether the larger effect on membrane potential during the trial epochs compared to quiet periods might be due to differences in the LC ISI distributions in these epochs (i.e. presence or absence of bursting rather than total spike counts).

2) Some justification or analysis of the specific time windows chosen for analysis of the pupil in Figure 2A would be useful. It is unclear why tone epochs were defined as [-0.2 0.3s] around tone onset. LC activity should not be affected in the period prior to tone as animals didn't know when the tone would occur.

3) Prestimulus whisking was found to be an important determinant of hit vs miss trials in a related whisker-detection paradigm (Kyriakatos et al., 2017). Does LC activity correlate with whisking, as well as licking? Here, the authors report that pre-whisker stimulus LC spiking differed across hit and miss trials (with lower spike rate in hit trials), suggesting the LC activity could participate in determining hit vs miss, but perhaps it is an indirect correlation because of whisking. This might be good to discuss, and perhaps analyze if whisker movements were tracked in these experiments. The authors might also discuss potential future optogenetic and pharmacological tests of the hypothesis that LC activity contributes to task performance.

4) The authors should give a more complete methods description. In general, I am not sure why the authors chose to write in such a short format, and they might expand on some points. How might the auditory stimulus drive LC spiking? How might LC spiking affect membrane potential in S1?

5) It is very intriguing that tone prior to stimulus presentation evoked strong LC responses but no changes in pupil size. The underlying mechanisms may be complicated, but some further discussion would be helpful. There may be an interesting link here with a recent work where task-evoked pupil dilation was decomposed to contributions of different cognitive processes, which may be differently modulated by the LC-NE system (Schriver et al., 2020).

6) In Figure 1E, the minimal pupil size seemed already fairly large. Was this panel a poor example or animals were already in a high arousal state throughout sessions?

7) The manuscript reported changes in raw pupil diameter in Figure 2, Figure 3, Figure 4. Given the large dynamic range of pupil fluctuations reported in literature, would Z-scored pupil size be a better way to lump data across animals/sessions?

8) It would be very interesting to know if there were any differences between regular spiking neurons and fast spiking neuron in terms of their correlation with LC spiking activity.

---

## [Author Response]

Essential revisions:1) Given the qualitative difference between the tonic and phasic modes in the LC, it would be helpful to have a plot like Figure 4—figure supplement 2, but conditioned on different inter-spike intervals. Specifically it would be interesting to know whether the larger effect on membrane potential during the trial epochs compared to quiet periods might be due to differences in the LC ISI distributions in these epochs (i.e. presence or absence of bursting rather than total spike counts).

We have done this analysis and now show the results as new panels in Figure 4—figure supplement 2. The results suggest that the ISI distribution may contribute, but cannot fully account for, the larger effect on V_m_ during trial epochs compared with quiet periods. Specifically, at the lowest spike count numbers, the ISIs for Tone/Lick epochs were lower than Quiet epochs, suggesting that there could be a partial role for ISI. However, although Quiet and Stimulus epochs differ for the V_m_ response across LC spike counts (Figure 4—figure supplement 2A, left panel), the ISIs are similar (Figure 4—figure supplement 2A, middle panel).

We now refer to this in the text:

“The relationship between V_m_ and LC spike count could partly but not fully be attributable to differences in the LC inter-spike intervals (Figure 4—figure supplement 2A).”

2) Some justification or analysis of the specific time windows chosen for analysis of the pupil in Figure 2A would be useful. It is unclear why tone epochs were defined as [-0.2 0.3s] around tone onset. LC activity should not be affected in the period prior to tone as animals didn't know when the tone would occur.

We initially chose a [-0.2 0.3 s] window for the Lick epochs, because LC activity could in principle either precede or follow lick movements. We then chose the same window for the other epochs for consistency. However, as noted by the Reviewers, this window need not precede the tone. We have therefore performed new analyses for the Tone epoch using a [0 0.5 s] window, and found that this did not significantly affect our conclusions. We show this new analysis in new Figure 4—figure supplement 3 and reference it in the Materials and methods:

“Task epochs were defined as: “Tone” epochs: -0.2 s to 0.3 s with respect to tone onset (we also tested 0 to 0.5 s, Figure 4—figure supplement 3)”

3) Prestimulus whisking was found to be an important determinant of hit vs miss trials in a related whisker-detection paradigm (Kyriakatos et al., 2017). Does LC activity correlate with whisking, as well as licking? Here, the authors report that pre-whisker stimulus LC spiking differed across hit and miss trials (with lower spike rate in hit trials), suggesting the LC activity could participate in determining hit vs miss, but perhaps it is an indirect correlation because of whisking. This might be good to discuss, and perhaps analyze if whisker movements were tracked in these experiments.

We do not have whisker tracking data for these experiments. However, we now discuss the possible link between whisking, LC activity and trial outcome:

“We found that pre-stimulus baseline LC spiking predicted behavioral responses, suggesting that being in a hyperactive or distractable state underlies failed stimulus detection. We excluded trials where mice licked before stimulus onset from our analysis, thus the higher baseline spiking in Miss trials was not caused by premature licking. However, it is possible that higher baseline LC activity was correlated with more whisking, which has been shown to lead to failed stimulus detection^25,26^.”The authors might also discuss potential future optogenetic and pharmacological tests of the hypothesis that LC activity contributes to task performance.

We have added to the Discussion as follows:

“Recent optogenetic and pharmacologic work has shown that activating the LC-NE system enhances detection/discrimination sensitivity^1,27,28^.”

And also:

“Future gain- and loss-of-function experiments are needed to determine causal roles of these distinct pathways to S1, and to determine how different LC activity patterns (i.e., tonic vs. phasic) affect sensory processing and task performance.”

4) The authors should give a more complete methods description. In general, I am not sure why the authors chose to write in such a short format, and they might expand on some points. How might the auditory stimulus drive LC spiking? How might LC spiking affect membrane potential in S1?

We have substantially expanded the Materials and methods section and now report several details here instead of only citing our past work. We agree this improves clarity.

We have added to the Discussion to address the other points. Specifically, we now write:

“We showed that S1 depolarizations immediately follow LC spiking, suggesting that LC activity at least partially contributes to the amount of depolarization in S1 neurons^4,5^. Given that sensory-evoked S1 responses predict the behavioral choices of the mice^21,23,30^, our results suggest that the LC-S1 pathway is involved during this task. However, the effect could be mediated directly through LC innervation of S1, or indirectly through LC innervating other early somatosensory areas, such as the VPM thalamus^1^. Future gain- and loss-of-function experiments are needed to determine causal roles of these distinct pathways to S1, and to determine how different LC activity patterns (i.e., tonic vs. phasic) affect sensory processing and task performance.”

We also now write:

“LC responded strongly to an auditory cue (tone) meant to alert the mice to the beginning of a trial. Such responses may be mediated via synaptic inputs to LC from the brainstem (e.g. nucleus paragigantocellularis)^31–34^.”

5) It is very intriguing that tone prior to stimulus presentation evoked strong LC responses but no changes in pupil size. The underlying mechanisms may be complicated, but some further discussion would be helpful. There may be an interesting link here with a recent work where task-evoked pupil dilation was decomposed to contributions of different cognitive processes, which may be differently modulated by the LC-NE system (Schriver et al., 2020).

We have added a paragraph to the Discussion to address these points. Specifically, we now write:

“Pupil size changes have been linked to activity in multiple brain areas and neuromodulatory systems^8,10,18^, and different pupil response profiles reflect different cognitive processes^37^. Therefore, it is possible that the pupil exhibits dynamic coupling with different underlying brain circuits in a cognitive process- (behavioral epoch-) dependent way. In addition, a recent study showed that pupil responses to dorsal raphe stimulation exhibited task uncertainty-dependent variations^18^. Thus, it is also possible that other modulatory systems (e.g., serotonergic and cholinergic)^30^ modulate the pupil-LC coupling in a dynamic manner.”

6) In Figure 1E, the minimal pupil size seemed already fairly large. Was this panel a poor example or animals were already in a high arousal state throughout sessions?

We now plot a histogram of median pupil size from all sessions, with the value of the example trace indicated (new Figure 1—figure supplement 1). This shows that the example trace is typical, and that the pupil in the example trace still has dynamic range for further dilation.

We are unable to use baseline pupil diameter to assess overall levels of arousal across mice and sessions, because we used controlled ambient light from an LED to put the pupil in a non-saturated state. However, our use of an auditory cue was intended to put the mice in as homogenous an arousal state as possible at the beginning of each trial. Our Discussion now refers to this fact in the following paragraph:

“However, it is possible that higher baseline LC activity was correlated with more whisking, which has been shown to lead to failed stimulus detection^25,26^. Overall, the effect was weak, possibly due to the use of an auditory cue that puts the mice in a more homogeneous arousal/attentive state. In other tasks without such alerting cues, task performance may have a stronger dependence on arousal and pre-stimulus LC activity.”

7) The manuscript reported changes in raw pupil diameter in Figure 2, Figure 3, Figure 4. Given the large dynamic range of pupil fluctuations reported in literature, would Z-scored pupil size be a better way to lump data across animals/sessions?

Several of our analyses will not be affected by z-scoring. These include the cross-correlation analyses (Figure 2B, Figure 3G-L, Figure 3—figure supplement 2A-B), the paired-observation analyses (Figure 2D and Figure 2—figure supplement 2A), and the analyses of peak timings shown in our histograms (Figure 3F and Figure 3—figure supplement 3B). For the remaining analyses, Z-scoring will scale the values for an individual recording, and potentially affect the degree to which the different recordings will contribute to averages. We have confirmed that z-scoring does not affect our conclusions for these remaining analyses. For the most critical analysis, that of Figure 4B, we have added a new Figure 4—figure supplement 1 to show a version of Figure 4B obtained after z-scoring the pupil data. The results are nearly identical. We now refer to this new Figure 4—figure supplement 1 as follows:

“In contrast, pupil dilation associated with an LC spike had the biggest response to licking and almost no response to the tone (Figure 4B; this was true as well for pupil dilation after z-scoring, Figure 4—figure supplement 1).”

8) It would be very interesting to know if there were any differences between regular spiking neurons and fast spiking neuron in terms of their correlation with LC spiking activity.

We agree this is an interesting question. However, we do not have identifiable fast-spiking neurons in our dataset of recordings.